# A Brain Network Analysis Model for Motion Sickness in Electric Vehicles Based on EEG and fNIRS Signal Fusion

**DOI:** 10.3390/s24206613

**Published:** 2024-10-14

**Authors:** Bin Ren, Pengyu Ren, Wenfa Luo, Jingze Xin

**Affiliations:** 1Shanghai Key Laboratory of Intelligent Manufacturing and Robotics, School of Mechatronic Engineering and Automation, Shanghai University, Shanghai 200444, China; renpengyu@shu.edu.cn; 2Zhejiang Key Laboratory of Robotics and Intelligent Manufacturing Equipment Technology, Ningbo Institute of Materials Technology & Engineering, Chinese Academy of Sciences, Ningbo 315201, China; 3SAIC Motor R&D Innovation Headquarters, SAIC Motor Corporation Limited, Shanghai 201804, China; luowenfa@immotors.com (W.L.); xinjingze@saicmotor.com (J.X.)

**Keywords:** EEG, fNIRS, sensor fusion, motion sickness recognition, brain network, GCN

## Abstract

Motion sickness is a common issue in electric vehicles, significantly impacting passenger comfort. This study aims to develop a functional brain network analysis model by integrating electroencephalography (EEG) and functional near-infrared spectroscopy (fNIRS) signals to evaluate motion sickness symptoms. During real-world testing with the Feifan F7 series of new energy-electric vehicles from SAIC Motor Corp, data were collected from 32 participants. The EEG signals were divided into four frequency bands: delta-range, theta-range, alpha-range, and beta-range, and brain oxygenation variation was calculated from the fNIRS signals. Functional connectivity between brain regions was measured to construct functional brain network models for motion sickness analysis. A motion sickness detection model was developed using a graph convolutional network (GCN) to integrate EEG and fNIRS data. Our results show significant differences in brain functional connectivity between participants in motion and non-motion sickness states. The model that combined fNIRS data with high-frequency EEG signals achieved the best performance, improving the F1 score by 11.4% compared to using EEG data alone and by 8.2% compared to using fNIRS data alone. These results highlight the effectiveness of integrating EEG and fNIRS signals using GCN for motion sickness detection. They demonstrate the model’s superiority over single-modality approaches, showcasing its potential for real-world applications in electric vehicles.

## 1. Introduction

As autonomous vehicle technology advances, the driver’s role increasingly becomes that of a passenger [1]. This shift makes passengers more prone to motion sickness than drivers, particularly in fully autonomous (Level 5) environments, where everyone in the vehicle may face the risk of motion sickness. Common physiological symptoms of motion sickness include cold sweats, dizziness, nausea, and vomiting. Monitoring these symptoms provides a reliable approach to studying motion sickness [2]. By continuously tracking changes in these physiological signals, the severity of motion sickness can be assessed, allowing for timely interventions to prevent the worsening of symptoms [3].

Brain imaging is a non-invasive technique used to capture data from the human brain and measure neural activity. Techniques currently in use include functional near-infrared spectroscopy (fNIRS), electroencephalography (EEG), and functional magnetic resonance imaging (fMRI). Researchers are increasingly leveraging these technologies to study motion sickness from the perspective of brain cognitive representations. Due to its high temporal resolution, EEG has become one of the most widely used tools [4]. Qin et al. processed -EEG signals before and after visual-induced motion sickness (VIMS) and identified statistically significant differences in the power spectral density (PSD) of δ, θ, and α bands within the C3 channel [5]. Yeo et al. found that the severity of motion sickness correlated with a significant increase in α and θ waves in the parietal and occipital lobes [6]. Li and colleagues developed a method based on wavelet packet transformation to extract energy ratios of δ, θ, α, and β rhythms from EEG signals, using K-nearest neighbor classifiers, polynomial kernel support vector machines, and radial basis function kernel support vector machines for the automatic detection of VR-induced motion sickness [7]. Bang et al. employed 28-channel EEG data combined with a convolutional neural network to create a motion sickness prediction model using dry EEG in a real driving environment [8]. These studies have utilized EEG signal analysis to uncover characteristics of motion sickness, revealing frequency band changes in different brain regions, and have developed machine learning-based models for the automatic detection and prediction of motion sickness. However, EEG is limited by low spatial resolution and is susceptible to environmental electromagnetic interference.

In recent years, fNIRS has rapidly emerged as a tool for monitoring brain activity. It provides insights into cerebral hemodynamics by measuring changes in oxygenated and deoxygenated hemoglobin in the cerebral cortex. fNIRS is widely used in research and clinical settings due to its low cost, safety, portability, and acceptable spatial resolution [9,10]. Compared to fMRI, fNIRS offers significant advantages in practical applications, including fewer restrictions on the experimental environment and greater robustness to motion artifacts, enhancing its potential for use in real-world scenarios [11]. Its effectiveness in studying motion sickness has been demonstrated in several studies. For example, Zhang et al. used fNIRS to discover that motion sickness significantly affects occipital lobe activity [12]. Kinoshita et al. measured the impact of 3D video on cerebral blood flow and found significant changes from the occipital lobe to the prefrontal cortex [13]. Additionally, Hoppes et al. investigated the effect of optic flow on brain activation, showing increased activation in the bilateral frontotemporal–parietal regions [14]. Ren’s team utilized a custom-built motion sickness simulation platform and fNIRS equipment to develop a model for assessing passengers’ motion sickness levels, validated by prefrontal cortex oxygenation signals and motion sickness scale scores. They successfully developed a classification model for identifying motion sickness symptoms by extracting features using principal component analysis and wavelet decomposition [15,16]. These studies utilized fNIRS to analyze the effects of motion sickness on different brain regions, revealing activation changes in the occipital lobe, prefrontal cortex, and bilateral frontotemporal-parietal regions, and successfully identified and assessed motion sickness symptoms through model development. However, fNIRS has poor temporal resolution due to the slow hemodynamic response [17].

Due to technological constraints and the intricate nature of neural processing, EEG, fNIRS, and fMRI each capture only a fragment of the brain’s functional data. In recent years, multimodal fusion technology has been widely applied in neuropathological diagnosis [18,19,20]. Within this context, integrating EEG and fNIRS has emerged as an efficient method for acquiring comprehensive brain data, with EEG capturing electrical brain activity and fNIRS monitoring hemodynamic responses. Early studies have demonstrated that the classification performance of EEG-fNIRS fusion surpasses that of single-modality approaches, often utilizing feature-level or decision-level fusion strategies to combine features and classification results [21,22]. Despite significant differences in their imaging principles and physiological bases, combining EEG and fNIRS can overcome their limitations, yielding higher-quality brain activation data and offering additional insights into neurovascular coupling (NVC) [23]. Numerous studies have confirmed that a bimodal EEG-fNIRS system provides richer feature information than single modalities, significantly improving classification accuracy and showing substantial potential in dynamic complex spatiotemporal processes [24,25,26,27]. However, research on the fusion of EEG and fNIRS in motion sickness still needs to be improved, representing a gap in the current literature.

The human brain is a complex network with higher-level functions depending on the coordinated interaction between different regions [28]. A functional brain network (FBN) is a model that describes the functional connections between various regions of the brain, conceptualizing the brain as a complex network composed of different regions (nodes) linked by functional connections (edges). These functional connections are quantified using channels’ coherence, correlation, and phase synchronization [29]. FBNs can uncover structural and functional brain characteristics—such as modularity, hierarchy, centrality, and hub distribution—that traditional analysis methods may miss. As a powerful analytical tool, FBNs are widely applied in fields like brain–computer interface classification, mental illness diagnosis, fatigue detection, and emotion cognition classification [30,31]. Recent research has begun to explore the effects of motion sickness on FBNs. For example, Toschi et al. used fMRI to show that motion sickness enhances neural connectivity between the visual motion processing areas and the regions involved in nausea [32]. Ruffle et al. also used fMRI to link nausea severity to specific functional networks, including the thalamus, various cingulate cortex regions, the caudate nucleus, and the nucleus accumbens [33]. Furthermore, Snodgrass et al. compared brain functional connectivity during nausea between healthy subjects and gastroparesis patients using dynamic visual stimulation, finding a significant reduction in connectivity between the right insula and the left temporal lobe in gastroparesis patients during nausea [34]. Nürnberg et al., employing EEG and VR technology, found that high-intensity visually induced motion sickness reduces information flow between brain regions [35]. Although these studies have revealed the impact of motion sickness and visually induced motion sickness on neural connectivity in specific brain regions, some limitations remain. Current research on motion sickness using EEG and fNIRS reveals two main shortcomings:Functional brain network studies related to motion sickness have predominantly utilized fMRI technology, while the application of EEG and fNIRS in this area has been relatively limited;Most studies have relied on either EEG or fNIRS alone without leveraging the combined strengths of both technologies.

Therefore, we used the weighted phase lag index (WPLI) to quantify the functional connectivity between EEG and fNIRS channels, constructing the corresponding functional brain network. This network comprises multiple nodes (representing different brain regions) and edges (representing functional connections) between them. As a typical complex network, the brain network exhibits unique properties that can be effectively represented and analyzed using graph theory. We modeled the brain network as a graph structure, with each channel corresponding to a node and the functional connectivity between node pairs represented as edges. This graph-based approach led us to employ a graph convolutional network (GCN) for modeling. The inherent graph structure of GCN is particularly suited to handling the multi-channel data and connectivity characteristics found in EEG and fNIRS, and previous studies have demonstrated that GCN excels in multimodal data fusion.

In this study, we first conducted a motion sickness induction experiment in a real vehicle setting, where we collected EEG and fNIRS signals from passengers and gathered motion sickness-related data through questionnaires. We obtained EEG and fNIRS signals under both motion sickness and non-motion sickness conditions, followed by preprocessing of the raw data. The EEG signals were divided into four sub-bands: δ, θ, α, and β. Changes in cerebral oxygen exchange (ΔCOE) values were derived through preprocessing for the fNIRS signals. We then constructed the functional brain networks for EEG by quantifying the functional connectivity between channels within the δ, θ, α, and β sub-bands. Similarly, we built the functional brain networks for fNIRS and analyzed the effects of motion sickness on these networks. Finally, we developed a motion sickness detection model based on the EEG and fNIRS functional brain networks. In this approach, the motion sickness detection task was formulated as a whole graph classification problem for an undirected graph. The graph nodes correspond to EEG or fNIRS channels, each represented by a feature vector. These node features and the adjacency matrix were then input into a graph convolutional network (GCN) to classify each graph with a motion sickness label. The primary contributions of this paper are as follows:Constructing functional brain networks using EEG and fNIRS to investigate the effects of motion sickness on brain functional connectivity;Developing an innovative motion sickness detection model by applying a graph convolutional network (GCN) to process EEG and fNIRS data;Conducting extensive experiments on real-vehicle datasets, demonstrating that the model achieves optimal performance when fNIRS signals are combined with EEG Beta band signals, resulting in an 8.2% improvement in the F1 score compared to using fNIRS alone and an 11.4% improvement compared to using the Beta band alone.

The remainder of this paper is structured as follows: Section 2 describes the construction process and analysis methods for the EEG and fNIRS functional brain networks and the datasets used. Section 3 details the specific architecture of the motion sickness detection model based on EEG-fNIRS multimodal data. Section 4 presents the results, focusing on the findings of this study. The results are explained and analyzed. Finally, Section 5 summarizes the main points and conclusions of this paper.

## 2. Functional Brain Network

This section will detail the construction process and analysis methods for the EEG and fNIRS functional brain networks and describe the datasets used. The logic and structure of this paper are illustrated in Figure 1.

### 2.1. Dataset

The dataset used in this paper was obtained from a collaborative study between Shanghai University and SAIC Motor Corporation, which collected passenger data during real-world driving conditions in electric vehicles. To capture EEG and fNIRS signals from participants experiencing motion sickness, we partnered with the SAIC Motor to conduct a series of real-vehicle tests, ensuring that data collection occurred in a genuine driving environment. After obtaining informed consent from the volunteers, 11 healthy participants were recruited from Shanghai University, consisting of 9 males and 2 females, aged between 22 and 26 years. Participants were first asked to complete the Motion Sickness Susceptibility Questionnaire (MSSQ) to assess their sensitivity to everyday motion sickness. The participants represented three levels of motion sickness sensitivity: non-susceptible, moderately susceptible, and highly susceptible. Including participants with varying sensitivity levels helped extract general patterns from the sample group.

All participants used the same EEG and fNIRS acquisition equipment with consistent settings during the experiment. The EEG data were collected using the Emotiv Epoc+, a portable 16-channel wireless monitoring device, while cerebral oxygenation was measured with the fNIRS-2000C near-infrared monitoring system from BIOPAC Systems, Inc., Goleta, CA, USA. As illustrated in Figure 2e, the EEG device featured 14 signal acquisition channels and 2 reference electrodes (CMS and DRL), with a sampling rate of 128 Hz. The 14 channels—AF3, F7, F3, FC5, T7, P7, O1, O2, P8, T8, FC6, F4, F8, and AF4—covered the frontal, temporal, parietal, and occipital regions of the brain. As shown in Figure 2f, the fNIRS device was equipped with 6 near-infrared light emitters and 2 receivers, forming 6 channels in total. All channels were positioned in the prefrontal cortex, labeled Channels 1 through 6. Channels 1 and 5 were located in the left prefrontal area, Channels 3 and 2 in the midline prefrontal area, and Channels 4 and 6 in the right prefrontal area, with a sampling rate of 10 Hz.

To capture motion sickness signals under real-world conditions, the experimental route was selected in Shanghai’s Baoshan District, following a specific loop that includes the Outer Ring Road, S7 Expressway, and Bao’an Road to Hutai Road, as depicted in Figure 2a. This route features several sharp curves and bumpy sections to induce motion sickness symptoms. The initial stage of the route consists of a flat straight section to collect baseline data from the volunteers. The route covers approximately 12.1 km, with a total driving time of 30 min. The experimental vehicle used was a Feifan F7 series new energy electric vehicle provided by SAIC Motor, as shown in Figure 2b. As illustrated in Figure 2c, two participants were seated in the vehicle’s left and right rear seats and equipped with EEG and fNIRS devices to collect data. An experiment recorder sat in the front passenger seat and asked the participants about their condition every 5 min, recording their scores using the Fast Motion Sickness (FMS) scale. The FMS scale used in this study is a 6-point Likert scale, as shown in Table 1, which allows for a quick assessment of the participants’ motion sickness levels during the experiment [15]. Subjects with an FMS score of 1 are classified as being in a non-motion sickness state, while those with a score greater than one are classified as in a motion sickness state. All participants underwent EEG and fNIRS signal acquisition under the same experimental conditions. Specifically, after EEG data collection, we allowed the subject to return to a resting state before collecting fNIRS data under the same experimental conditions. This procedure ensured that each EEG and fNIRS dataset originated from the same subject under consistent experimental settings. In total, 32 experimental sessions were completed.

To ensure the data reflect physiological changes associated with motion sickness and not artifacts or external interference, we strictly controlled experimental conditions and implemented rigorous preprocessing methods. These steps effectively reduced potential data contamination, ensuring accurate and reliable analysis. EEG captures the electrical activity of synchronized neurons through scalp electrodes, recording waveforms to monitor brain function. However, it is susceptible to artifacts from physiological sources (e.g., eye movements, muscle activity) and non-physiological sources (e.g., environmental interference). The data are typically filtered between 0.5 Hz and 45 Hz to enhance signal accuracy and reduce noise, followed by Independent Component Analysis (ICA) to remove motion artifacts [36]. Using a Butterworth band-pass filter, the EEG signals are then decomposed into four sub-bands—δ (0–4 Hz), θ (4–8 Hz), α (8–12 Hz), and β (12–32 Hz) [37]. Similarly, fNIRS measures brain oxygenation by tracking changes in near-infrared light absorption but is also influenced by physiological and environmental noise. Preprocessing first involves a visual inspection to remove segments with obvious localized interference, followed by the application of a bandpass filter to eliminate high-frequency noise. Motion artifacts are then addressed using the SMAR algorithm, and the light intensity data are converted into changes in oxyhemoglobin (HbO) and deoxyhemoglobin (HbR) concentrations. Finally, cerebral oxygen exchange (ΔCOE) is calculated, with ΔCOE serving as a measure of brain oxygenation. The cerebral oxygen exchange (ΔCOE) is then calculated using Formula (1) [12]. ΔCOE quantifies brain oxygenation status, and in motion sickness studies, changes in ΔCOE provide insights into oxygen exchange variations linked to neural activity.
(1)$$∆COE=\frac{∆C_{HbR}-∆C_{HbO}}{\sqrt{2}}$$
where $∆C_{HbR}$ and $∆C_{HbO}$ represent the changes in deoxygenated hemoglobin and oxygenated hemoglobin concentrations, respectively.

This study compiled a dataset for motion sickness analysis using preprocessed EEG and cerebral oxygen exchange (COE) data combined with subjective motion sickness assessment scores. The dataset consists of recordings from 32 subjects, with 32 EEG and 32 sets of fNIRS data. Each experiment lasted 30 min, and FMS scores were recorded every 5 min. As a result, each subject generated 6 5-min segments of raw EEG and fNIRS data, resulting in a total of 192 segments across all subjects. After data preprocessing, 136 valid 5-min EEG and fNIRS segments were retained. Based on the FMS scores, segments with a score of 1 were categorized as representing a “non-motion sickness” state, while segments with scores greater than one were classified as “motion sickness.” Ultimately, the dataset includes 61 segments corresponding to the non-motion sickness state and 75 segments for the motion sickness state.

### 2.2. Network Construction

This study constructs functional brain networks using EEG and fNIRS data collected from the dataset. For the EEG data, brain networks were constructed for each frequency band, focusing on four classic bands: δ (0–4 Hz), θ (4–8 Hz), α (8–12 Hz), and β (12–32 Hz). For the fNIRS data, the brain network was constructed using ΔCOE data. The construction process for these functional brain networks is detailed in Figure 3 and involves several key steps. First, based on the study’s objectives, brain regions or corresponding device channels are selected as network nodes, representing specific brain areas and serving as the basic units for functional analysis. Next, time series data are extracted from the selected nodes, reflecting their activity during the experiment. Subsequently, functional connectivity between nodes is quantified. Finally, an appropriate threshold is applied to determine which connections between nodes are strong enough to be included in the functional brain network [38].

In this process, defining the nodes and edges of the brain network is a critical step. Each channel corresponds to a different brain region for EEG and fNIRS multi-channel signal acquisition devices, so each channel is typically defined as a node. A functional brain network is constructed by quantifying the relationships between signals from different channels and using the strength of these relationships to represent the functional connectivity between corresponding brain regions. In the EEG functional brain network, 14 nodes correspond to the 14 channels of the EEG device, while in the fNIRS functional brain network, 6 nodes correspond to the 6 channels of the fNIRS device. To quantify the strength of functional connectivity between channel signals, the Weighted Phase Lag Index (WPLI) is employed [39]. WPLI measures phase coupling in brain signals, focusing on non-synchronous phase differences. By weighting phase differences across various frequencies, WPLI provides a more accurate representation of the connectivity between different brain regions. The formula for calculating WPLI is as follows:(2)$$WPLI=\frac{\left|\langle I\left(S\right)\rangle \right|}{\langle \left|S\right|\rangle }=\frac{\left|\langle \left|I\left(S\right)\right|sign\left(I\left(S\right)\right)\rangle \right|}{\langle \left|I\left(S\right)\right|\rangle }$$

In this context, let $X\left(t\right)$ and $Y\left(t\right)$ represent the preprocessed time series signals from two channels, with S being the cross-spectrum of $X\left(t\right)$ and $Y\left(t\right)$. The WPLI ranges from 0 to 1, where higher values indicate a stronger coupling of neural oscillatory activity and lower values indicate weaker coupling.

After obtaining the adjacency matrix representing the relationships between channels, an appropriate threshold is set to convert it into a binary matrix. If the connection strength between nodes exceeds the threshold, it is set to 1, indicating a connection; otherwise, it is set to 0. A higher threshold results in fewer edges, potentially overlooking important connections, while a lower threshold results in more edges, potentially introducing false connections. Thus, the threshold setting directly affects the network’s topology and properties. This study used two constraints to determine the threshold: (1) ensuring that the network’s average node degree is not lower than the natural logarithm of the number of nodes N and (2) ensuring that the network exhibits small-world characteristics [40]. Based on these considerations, the thresholds for the δ, θ, α, and β bands were set at 0.36, 0.42, 0.40, and 0.26, respectively, while the threshold for fNIRS was set at 0.35. These criteria were used to construct the functional brain networks for EEG and fNIRS.

Theoretically, a network can be represented as a graph consisting of vertices (nodes) and edges (connections). The brain can be simplified through graph theory into a set of vertices and edges, where the presence or absence of an edge reflects the importance or strength of the interaction between two nodes. Research indicates that using graph theory to analyze brain networks’ topological properties and dynamic evolution is practical [41]. This study used graph theory to extract attribute metrics from EEG and fNIRS brain networks. We analyzed their association with motion sickness to explore changes in functional connectivity between brain regions during motion sickness. The primary metrics include node degree, clustering coefficient, and local and global efficiency. We then applied the Mann–Whitney test to assess differences in these parameters between non-motion sickness and motion sickness states, setting the significance level threshold at 0.05.

The node degree is the most fundamental metric in a network. The node degree, $D_{i}$, is defined as the number of nodes directly connected to node i within the network G. A higher node degree indicates that the node has more connections, making it more significant within the brain network. The formula for calculating the node degree $D_{i}$ is as follows:(3)$$D_{i}=\sum_{j\in G}a_{ij}$$
where $a_{ij}$ represents the edge that connects node i to node *j* within the network G.

The clustering coefficient measures the clustering characteristics and cohesion within a brain network, indicating the likelihood that the neighbors of a given node are also connected. For node i, the clustering coefficient is defined as the ratio of the actual number of edges $E_{i}$ between its neighboring nodes to the maximum possible number of edges among those neighbors. The formula for calculating the clustering coefficient is as follows:(4)$$C_{i}=\frac{2E_{i}}{k_{i}\left(k_{i}-1\right)}$$
where $k_{i}$ represents the number of neighboring nodes that node i has and $k_{i}\left(k_{i}-1\right)/2$ represents the maximum possible number of edges among these $k_{i}$ neighboring nodes. Due to the large number of nodes in complex networks, we do not examine the clustering coefficient of each i node in the brain network. Instead, we analyze the average clustering coefficient of the entire network. In an unweighted network, the average clustering coefficient $C_{i}$ is calculated as the average of the clustering coefficients of all nodes, reflecting the clustering connections around individual nodes. The calculation formula is as follows:(5)$$C_{i}=\frac{1}{N}\sum_{i=1}^{N}C_{i}$$

The global efficiency $E_{glop}\left(G\right)$ of a brain network reflects the network’s overall capacity for information transmission. The shorter the shortest path length between nodes, the faster the information transfer and the higher the global efficiency. The formula for calculating the global efficiency is as follows:(6)$$E_{glop}\left(G\right)=\frac{1}{N}\sum_{i\in G}E_{glob}\left(i\right)$$

The local efficiency $E_{loc}\left(G\right)$ of a brain network reflects its capacity for local information transmission.
(7)$$E_{loc}\left(G\right)=\frac{1}{N}\sum_{i\in G}E_{loc}\left(i\right)$$

## 3. Recognition Model Based on the Fusion of EEG and fNIRS

Given the topological structure of brain networks, we define motion sickness recognition based on these networks as a whole-graph classification problem. The motion sickness recognition model is illustrated in Figure 4. First, brain networks are constructed from EEG and fNIRS datasets. Each subject’s EEG and fNIRS brain networks are then converted into undirected graphs $G=\left(V,\;E\right)$, where G consists of n nodes V and $C_{n}^{2}$ edges. In the EEG undirected graph, each node $v_{i}\;\in \;V$ corresponds to an EEG channel and has a feature vector $f_{i}\;\in \;R^{d_{0}}$ with $d_{0}$ dimensions. Similarly, in the fNIRS undirected graph, each node $v_{i}\;\in \;V$ corresponds to an fNIRS channel, also represented by a feature vector $f_{i}\;\in \;R^{d_{0}}$. In addition to calculating four graph-theoretic features, we include time–frequency domain features from the channel data, as detailed in Table 2. This approach allows for a more comprehensive analysis of node attributes and behavior. Each node has 13 features, so the feature dimension $d_{0}$ is 13. The graph feature matrix for each subject is therefore $X\;\in \;R^{n\times d_{0}}$. An edge $\left(v_{i},\;e,v_{j}\right)$ exists between any two random nodes $v_{i}$ and $v_{j}$, where e represents the channel connectivity, specifically the functional connectivity coefficient of each channel. We construct the graph’s adjacency matrix $A\;\in \;R^{n\times n}$ by setting an appropriate threshold to select channels with effective connections. Finally, the node feature matrix X and the adjacency matrix A are input into the model, with a classification label $y\in \left\{0,\;1\right\}$ assigned to each graph $G=\left(V,\;E\right)$, where label 0 indicates a non-motion sickness state and label 1 indicates a motion sickness state.

This study developed a GCN architecture consisting of four graph convolutional layers, two pooling layers, and one fully connected layer to maintain the integrity of node features and reduce information loss, as shown in Figure 4. The first two graph convolutional layers are designed to extract node features and adjacency matrices from the undirected EEG and fNIRS graphs. These layers aim to capture independent features from each modality. The third graph convolutional layer integrates the features from EEG and fNIRS extracted by the previous layers. A bimodal adjacency matrix is constructed based on the EEG and fNIRS functional connectivity networks and their spatial topological relationship. This matrix bridges the two modalities by adding connections with a strength of one at overlapping channels without altering the individual networks. The feature-length is increased from 13 to 30 in the first and second layers to enhance node feature representation and from 30 to 60 in the third layer. A TopKPooling layer then reduces the number of nodes from 20 to 10, followed by a fourth graph convolutional layer that increases the feature-length to 90. Another TopKPooling layer reduces the node count to 1, resulting in a graph with a single node and 90 features. This 90-dimensional feature vector is passed through a fully connected layer, producing a two-dimensional vector for classification. The core of this architecture is the effective mapping of EEG and fNIRS data into a graph structure, aligning this information with the subject’s subjective evaluations to accurately identify the motion sickness state.

This study also evaluated the performance of motion sickness detection using EEG and fNIRS signals individually to assess the effectiveness of the GCN in feature fusion. During model training, cross-entropy was used as the loss function, the Adam optimizer was employed for parameter optimization, and L2 regularization was introduced to prevent overfitting. We combined grid search with manual tuning to determine the model’s hyperparameters accurately. Initially, a coarse grid search was performed to narrow the range of possible hyperparameters. This was followed by 10-fold cross-validation to validate each parameter combination’s effectiveness, ensuring the evaluation results’ stability and reliability. Stratified sampling was used during dataset splitting to maintain consistent class proportions. The study focused primarily on accuracy, binary F1 score, recall, and precision. The F1 score is especially critical in motion sickness detection because it balances precision and recall, offering a comprehensive measure of detection accuracy. Given the differing costs of false negatives and false positives, the F1 score provides a more meaningful evaluation. Therefore, we emphasized the F1 score as the primary evaluation metric. For each set of hyperparameters, we recorded the average performance across all dataset splits to evaluate the effectiveness of that set. Ultimately, the hyperparameter set that achieved the highest F1 score was selected as the optimal configuration for each method.

## 4. Result

### 4.1. EEG and fNIRS Brain Networks

Figure 5a presents the average WPLI matrix constructed from the fNIRS data of all subjects in both non-motion and motion sickness states. The results show that, except for Channels 4 and 6, the WPLI values of all other channels increase during motion sickness. The WPLI matrix is converted into the six-channel fNIRS functional brain network shown in Figure 5b by applying a threshold. These results suggest that motion sickness alters the structure of the prefrontal functional brain network. In the non-motion sickness state, the fNIRS brain network is relatively sparse; however, the connectivity within the fNIRS brain network is enhanced during motion sickness. Key nodes in the network were identified by calculating normalized betweenness centrality and are marked in red in Figure 5b. In the non-motion sickness state, Channel 5 is the critical node, while in the motion sickness state, the key nodes shift to Channels 4, 2, and 5. This indicates that in the non-motion sickness state, the left frontal lobe has more connections, whereas during motion sickness, connectivity increases across the entire prefrontal cortex, with a significant enhancement in the right frontal lobe. Figure 5c illustrates the node degree distribution of the fNIRS brain network. During motion sickness, the node degree of all channels increases. Figure 5d compares the clustering coefficient, global efficiency, and local efficiency of the fNIRS brain network in both states. The clustering coefficient, global efficiency, and local efficiency are significantly enhanced during motion sickness. Table 3 provides the calculated values and statistical test results for the global efficiency, clustering coefficient, and local efficiency of the fNIRS brain network, all showing statistically significant differences.

Figure 6 displays the average WPLI matrices constructed from EEG data in the four sub-bands (δ, θ, α, and β) for all subjects in both non-motion sickness and motion sickness states. The results show that each sub-band’s brain network exhibits different patterns in the two states. By applying the set thresholds, the WPLI matrices for the four sub-bands were converted into adjacency matrices to construct the brain networks, as shown in Figure 6. Similarly, important nodes were identified by calculating the normalized betweenness centrality, marked in red in Figure 7. In the δ band brain network, the key nodes change between non-motion and motion sickness states. In the non-motion sickness state, the key nodes are AF3, AF4, T7, and F8, while in the motion sickness state, they are F4, F8, and P8. This indicates that the frontal and left temporal regions have more connections in the non-motion sickness state. In contrast, the right frontal and right parietal regions have more connections in the motion sickness state. For the θ band brain network, the key nodes in the non-motion sickness state are AF4, F8, T7, and T8, while in the motion sickness state, they shift to F3, F7, F8, FC5, and T8. Thus, the frontal and temporal regions have more connections in the non-motion sickness state, whereas in the motion sickness state, the frontal and right temporal regions have more connections. In the α band brain network, the transition from non-motion sickness to motion sickness also involves changes in key nodes. In the non-motion sickness state, the key nodes are F3, F4, F7, F8, T7, T8, and O2, while in the motion sickness state, they become F3, AF4, F8, FC5, T8, and O1. Therefore, in the non-motion sickness state, the frontal, temporal, and right occipital regions have more connections, while in the motion sickness state, the frontal, right temporal, and left occipital regions have more connections. In the β band brain network, the key nodes in the non-motion sickness state are AF4, F4, F8, T8, P7, and O2, while in the motion sickness state, they are F4, F8, FC5, and T8. Thus, the right frontal, right temporal, left parietal, and right occipital regions have more connections in the non-motion sickness state. In contrast, in the motion sickness state, the right frontal and right temporal regions have more connections.

Figure 8a illustrates the degree distribution of brain networks across the δ, θ, α, and β frequency bands. In the δ band during motion sickness, the node degrees of AF3, F7, O2, and T7 decrease, while those of F3, FC5, O1, P8, T8, and F4 increase. The degrees of nodes P7 and FC6 remain unchanged, as does F8, while AF4 decreases. In the θ band during motion sickness, the degrees of nodes AF3, T7, P7, O2, FC6, and F8 decrease, whereas F7, F3, FC5, O1, P8, and F4 increase; T8 and AF4 remain unchanged. In the α band, the degrees of nodes AF3, F7, T7, P7, O1, O2, P8, FC6, F4, and F8 decrease, with no change observed in F3, FC5, and T8. In the β band during motion sickness, there is a decrease in the degree of all nodes: AF3, F7, F3, FC5, T7, P7, O1, O2, P8, T8, FC6, F4, F8, and AF4. Figure 8b presents the clustering coefficient, global efficiency, and local efficiency of brain networks in the δ, θ, α, and β bands under both non-motion sickness and motion sickness conditions. Regarding global efficiency, all bands except the δ band exhibit a decrease during motion sickness, with the β band showing the most pronounced decline. Clustering coefficient analysis indicates a reduction across all bands except the δ band. Similarly, local efficiency analysis reveals a decrease in all bands except the δ band. Table 3 provides a summary of the computed global efficiency, clustering coefficient, and local efficiency for the δ, θ, α, and β bands, along with the results of statistical tests. Notably, the brain network in the β band shows statistically significant differences in global efficiency, clustering coefficient, and local efficiency.

### 4.2. The Analysis of Motion Sickness Recognition

Table 4 details the performance of the motion sickness recognition model across different brain network configurations, using both unimodal and multimodal input approaches. The model used fNIRS or individual EEG frequency bands (Delta, Theta, Alpha, Beta) as input in the unimodal setup. The model combined fNIRS data with each EEG frequency band for multimodal input. In the unimodal input, fNIRS achieved an F1 score of 0.766, while the F1 scores for EEG varied by frequency band, with the Delta band scoring the lowest at 0.693 and the Theta band the highest at 0.781. In the multimodal setup, combining fNIRS with EEG frequency bands generally improved performance, especially the combination of fNIRS and the Beta band, which achieved an F1 score of 0.848. These results indicate that fusing EEG and fNIRS signals enhances the model’s performance.

## 5. Conclusions

In this study, we analyzed the specific impact of motion sickness on functional brain networks by collecting fNIRS and EEG signals from passengers in real-world conditions. The findings revealed a significant enhancement in functional connectivity between the frontal and temporal lobes during motion sickness, particularly within the δ, θ, α, and β frequency bands and the fNIRS network structure. This change suggests that the brain’s functional connectivity undergoes reorganization to cope with the physiological and psychological stress induced by motion sickness. Additionally, we developed a motion sickness recognition model based on a graph convolutional network, which significantly improved diagnostic accuracy by leveraging the multimodal characteristics of EEG and fNIRS signals. The model showed the most significant performance improvement when combining fNIRS signals with the EEG β band, highlighting the advantages of this multimodal approach in capturing brain function changes. Overall, these findings deepen our understanding of the neural mechanisms underlying motion sickness and provide new insights and methodologies for diagnosing and intervening in motion sickness. Future research could explore changes in functional connectivity in other brain regions and investigate dynamic brain functions under different motion sickness conditions, offering more precise targets for treatment. Moreover, the innovative model developed in this study demonstrates the immense potential of multimodal data fusion technology in diagnosing neurological diseases, indicating promising directions for the future development of diagnostic technologies.

## Figures and Tables

**Figure 1 sensors-24-06613-f001:**
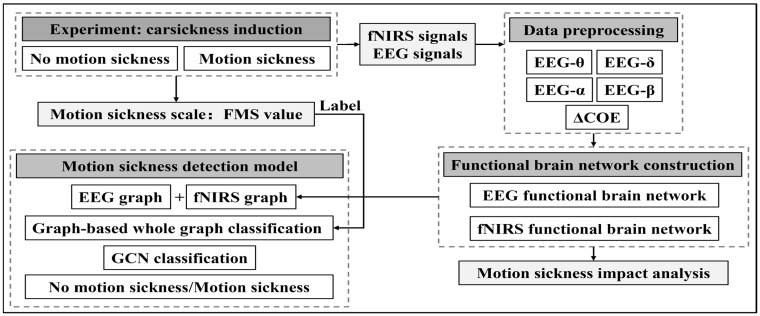
Logic and structure of this study.

**Figure 2 sensors-24-06613-f002:**
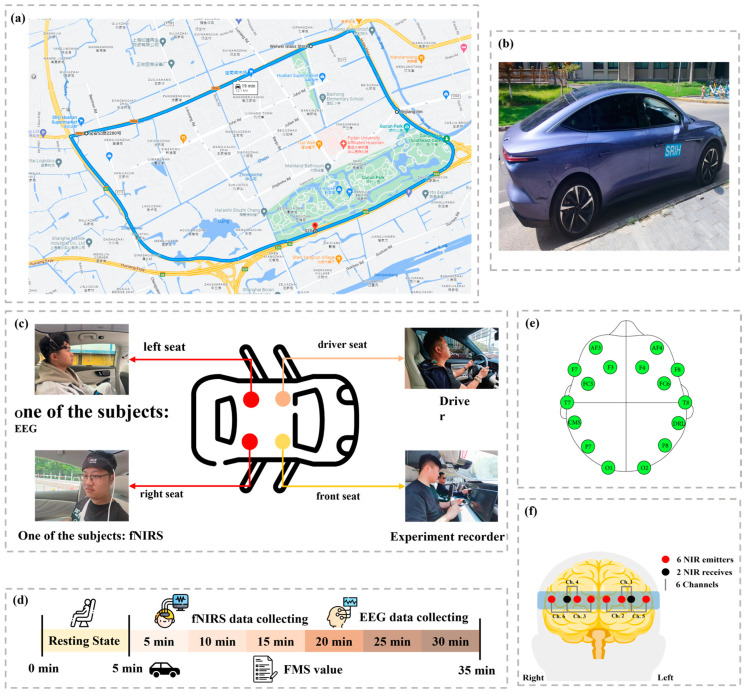
Details of our data collection process are as described below. (**a**) The electric vehicle used in the experiment. (**b**) The layout of the experimental route. (**c**) The seating arrangement of participants and the experimenter during the experiment. (**d**) The experimental procedure. (**e**) The EEG device and its electrode placement. The EEG device follows the International 10–20 system for channel distribution, including 14 channels: AF3, F7, F3, FC5, T7, P7, O1, O2, P8, T8, FC6, F4, F8, and AF4. Additionally, there are 2 reference electrodes, CMS and DRL. This EEG setup covers the frontal, temporal, parietal, and occipital lobes. (**f**) The fNIRS device configuration. The fNIRS system includes 6 near-infrared light emitters and 2 near-infrared receivers, with each emitter–receiver pair forming a channel, totaling 6 channels. This setup primarily covers the prefrontal cortex.

**Figure 3 sensors-24-06613-f003:**
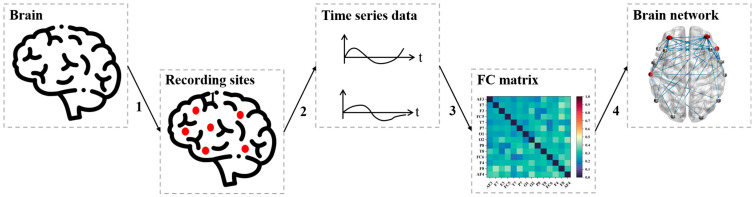
Flow chart of the functional brain network construction in this study.

**Figure 4 sensors-24-06613-f004:**
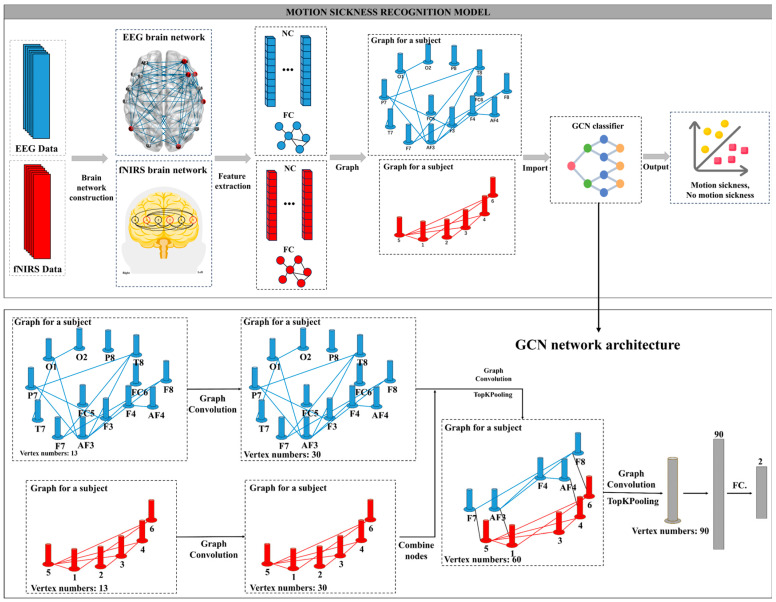
Motion sickness detection model. NC = node features, FC = functional connectivity. Among them, AF3, F7, F3, FC5, T7, P7, O1, O2, P8, T8, FC6, F4, F8, and AF4 are EEG channels. Channels 1, 2, 3, 4, 5, and 6 are fNIRS channels.

**Figure 5 sensors-24-06613-f005:**
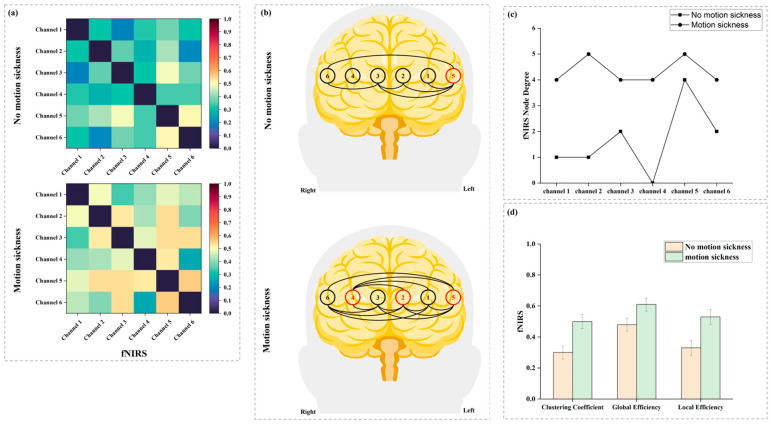
This figure illustrates the average WPLI matrix, functional brain networks, and their properties, constructed from all subjects’ functional near-infrared spectroscopy (fNIRS) data in both non-motion and motion sickness states. (**a**) WPLI matrices in non-motion sickness and motion sickness states. These 6 × 6 symmetric correlation matrices have the X and Y axes representing the corresponding fNIRS channels. To eliminate autocorrelation effects, the WPLI values on the diagonal (from the top left to the bottom right) are set to 0. (**b**) fNIRS functional brain networks in non-motion sickness and motion sickness states. The six nodes in the network correspond to the six channels of the fNIRS device, distributed across the prefrontal cortex, with red markers indicating key nodes within the brain network. (**c**) Node degree of the fNIRS functional brain networks in non-motion and motion sickness states. (**d**) Average clustering coefficient, global efficiency, and local efficiency of the fNIRS functional brain networks in non-motion sickness and motion sickness states.

**Figure 6 sensors-24-06613-f006:**
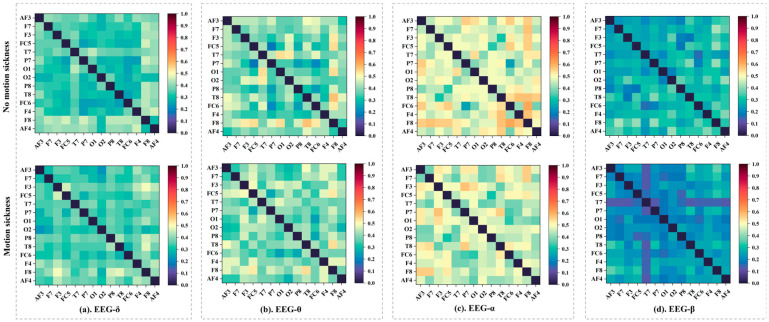
WPLI matrices constructed from the four sub-bands of 14-channel EEG data in non-motion sickness and motion sickness states.

**Figure 7 sensors-24-06613-f007:**
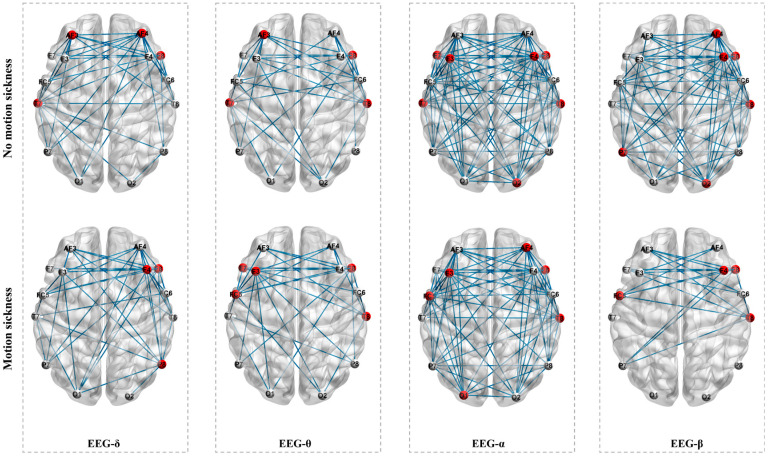
Brain networks constructed from the four sub-bands of 14-channel EEG data in non-motion sickness and motion sickness states. The 14 nodes correspond to the 14 EEG channels distributed across the brain’s frontal, temporal, parietal, and occipital regions. Red indicates the key nodes within the brain network.

**Figure 8 sensors-24-06613-f008:**
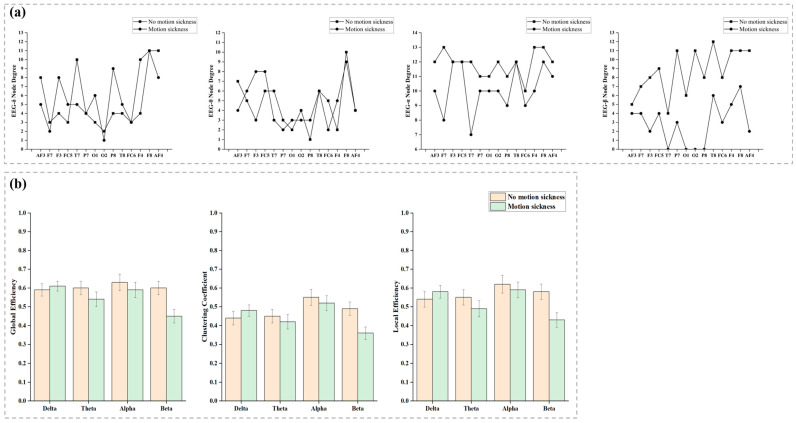
The attribute parameters of the brain network in the δ, θ, α, and β frequency bands are analyzed in this section. Panel (**a**) illustrates the changes in node degree for each channel within the brain network during non-motion sickness and motion sickness states. Panel (**b**) depicts the variations in the brain network’s average clustering coefficient, global efficiency, and local efficiency under both non-motion and motion sickness conditions.

**Table 1 sensors-24-06613-t001:** Fast Movement Sickness Scale.

Score	Description
1	No motion sickness at all: participants have no sensation of motion sickness and feel normal.
2	Slight motion sickness: participants have slight motion sickness but can still carry on with normal activities.
3	Some motion sickness: participants experience motion sickness symptoms and may feel discomfort but can still carry on with normal activities.
4	Much motion sickness: participants experience moderate motion sickness symptoms such as dizziness and nausea, which affect their ability to carry on with normal activities.
5	Extreme motion sickness: participants experience severe motion sickness symptoms such as severe dizziness and nausea, which prevent them from carrying on with normal activities.
6	Unbearable motion sickness: participants experience extremely severe motion sickness symptoms such as unbearable dizziness, nausea, and vomiting, which require immediate cessation of the activity and medical attention.

**Table 2 sensors-24-06613-t002:** Node features.

Feature	Description
MAV	The mean absolute value is often used to describe the average amplitude of a signal. It calculates the average of the signal’s absolute values, which can indicate the magnitude of a periodic waveform.
VAR	Variance measures the degree of dispersion in a data distribution. A higher variance indicates a more dispersed distribution. In signal processing, variance can be used to quantify the signal’s energy or the magnitude of its fluctuations.
Max	The highest point of the signal during the measurement period.
Min	The lowest point of the signal during the measurement period.
RMS	A measure of signal strength, typically used with periodic waveforms, calculated as the square root of the mean of the squares.
Kurtosis	An indicator that measures the shape of the signal probability distribution, reflecting the frequency and degree of signal extreme values.
Skewness	A measure of the asymmetry of the signal distribution. Positive skew means the tail of the distribution extends to the right; negative skew means it goes to the left.
PSD	Power spectral density describes the distribution of a signal’s power across different frequencies. It provides detailed information about the frequency content of the signal and is a key tool for analyzing frequency domain characteristics in signal processing.
Degree	Node degree is the most basic and essential metric in a network. A higher node degree indicates that the node has more connections, making it more integral and significant within the brain network.
Clustering coefficient	The clustering coefficient assesses the clustering properties and cohesion within a brain network, indicating the likelihood that a node’s neighbors are also connected to each other.
Global efficiency	Global efficiency reflects the overall capacity of a network to transmit information. The shorter the shortest path length between nodes, the faster the information is transmitted, leading to higher global efficiency.
Local efficiency	Local efficiency reflects the capacity for information transmission within local regions of the network.

**Table 3 sensors-24-06613-t003:** Functional brain network parameters for each sub-band in NM and MS states.

Data	Clustering Coefficient			Global Efficiency			Local Efficiency		
	NM	MS	*p*	NM	MS	*p*	NM	MS	*p*
fNIRS	0.30 ± 0.043	0.50 ± 0.046	**	0.48 ± 0.041	0.61 ± 0.042	***	0.33 ± 0.048	0.53 ± 0.049	***
Delta	0.44 ± 0.036	0.48 ± 0.030		0.59 ± 0.033	0.61 ± 0.026		0.54 ± 0.041	0.58 ± 0.034	
Theta	0.45 ± 0.036	0.42 ± 0.038		0.60 ± 0.036	0.54 ± 0.039		0.55 ± 0.041	0.49 ± 0.043	
Alpha	0.55 ± 0.043	0.52 ± 0.039		0.63 ± 0.044	0.59 ± 0.041		0.62 ± 0.047	0.59 ± 0.042	
Beta	0.49 ± 0.036	0.36 ± 0.033	**	0.48 ± 0.041	0.61 ± 0.042	**	0.33 ± 0.048	0.53 ± 0.049	**

Note: Data are the mean ± variance values of the parameters. The statistical significance of the differences between the non-motion sickness and motion sickness conditions (Mann–Whitney test, **: *p* < 0.01; ***: *p* < 0.001). NM = no motion sickness, MS = motion sickness.

**Table 4 sensors-24-06613-t004:** Performance of the motion sickness recognition model.

Brain Network Structure		Accuracy	F1 Score	Recall	Precision
Baseline (Single mode)	fNIRS	0.714	0.766	0.852	0.719
	Delta	0.675	0.693	0.695	0.755
	Theta	0.742	0.781	0.829	0.763
	Alpha	0.757	0.774	0.761	0.826
	Beta	0.700	0.734	0.788	0.728
Ours (Multimodality)	fNIRS + Delta	0.780	0.801	0.814	0.822
	fNIRS + Theta	0.780	0.798	0.813	0.795
	fNIRS + Alpha	0.780	0.816	0.888	0.785
	fNIRS + Beta	0.824	0.848	0.886	0.832

## Data Availability

All data included in this study are available upon request by contact with the corresponding author. The data are not publicly available because of ethical restrictions.

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
