# Peer review of "A Brain Network Analysis Model for Motion Sickness in Electric Vehicles Based on EEG and fNIRS Signal Fusion"

_sensors, 2024, doi:10.3390/s24206613_

Round 1
Reviewer 1 Report
Comments and Suggestions for Authors
- The reliability of the corresponding score of motion sickness reported is a key concern of this study.
- Another important issue that the authors need to consider and validate is whether the EEG and fNIRS data were not contaminated and dominated by another physiological state rather than the effect of motion sickness.
- It is not clear whether the EEG and fNIRS data were recorded from the same person when the data were applied to the computational analysis.
- The quantitative features used in this study are not novel.
- Furthermore, it needs to be verified how those selected features perform compared to the selected common features.
- The authors must validate and examine the method and data corresponding to the associated score of motion sickness and the three levels.
Author Response
Dear Reviewer,
First and foremost, we would like to express our sincere gratitude for your thorough review and valuable comments and suggestions on our manuscript. We have carefully read your feedback and made corresponding revisions to the paper to address your concerns. Below are our responses and solutions to each issue raised in your review.
Comment 1: The reliability of the corresponding score of motion sickness reported is a key concern of this study.
Response: Thank you for your valuable comments on our manuscript. We have carefully considered your suggestions and made the following revisions:
This study used the Fast Motion Sickness Scale (FMS) to obtain participants' corresponding motion sickness scores, enabling a quantitative assessment of motion sickness severity [1]. The FMS is widely used in motion sickness research, and its reliability has been well-established. The version of the FMS we adopted is a 6-point Likert scale, which allows for a quick assessment of participants' motion sickness levels during the experiment.
To further enhance the reliability of the scores, we inquired about the participants’ physical condition at regular intervals (every 5 minutes) during the experiment. We recorded their scores based on their immediate perceptions. Each participant’s score was evaluated using a standardized process, closely monitored by the experiment recorder. Additionally, recognizing that motion sickness is a subjective experience, we minimized bias by ensuring all participants were tested under the same experimental conditions. We also conducted a pre-study assessment using the Motion Sickness Susceptibility Questionnaire (MSSQ) to evaluate the participants' sensitivity to motion sickness, ensuring the consistency and reproducibility of the experimental results [2]. Furthermore, to ensure the representativeness of the data, we included participants with varying levels of motion sickness sensitivity (low, moderate, and high) to capture a wide range of individual differences.
Lastly, the collected EEG and fNIRS data were segmented into 5-minute intervals and matched with the corresponding FMS scores. When the FMS score was 1, we classified the participant as being in a non-motion sickness state, and when the score was greater than 1, we classified the participant as being in a motion sickness state. This method allowed us to categorize the EEG and fNIRS data effectively. We have supplemented this explanation in the revised manuscript.
We hope these changes have addressed your concerns and improved the clarity of our manuscript. Thank you again for your helpful feedback.
[1] Ren B, Guan W, Zhou Q. Study of Motion Sickness Model Based on fNIRS Multiband Features during Car Rides[J]. Diagnostics, 2023, 13(8): 1462.
[2] Ren B, Zhou Q. Assessing Passengers’ Motion Sickness Levels Based on Cerebral Blood Oxygen Signals and Simulation of Actual Ride Sensation[J]. Diagnostics, 2023, 13(8): 1403.
Comment 2: Another important issue that the authors need to consider and validate is whether the EEG and fNIRS data were not contaminated and dominated by another physiological state rather than the effect of motion sickness.
Response: Thank you for your insightful comment regarding the potential contamination of EEG and fNIRS data by other physiological states.
We recognize the importance of ensuring that the recorded data reflects motion sickness rather than being influenced by other physiological states. To address this concern, we implemented several data preprocessing steps to minimize the influence of unrelated physiological artifacts:
- Artifact Removal in EEG Data: We employed Independent Component Analysis (ICA) to isolate and remove common artifacts such as eye movements (e.g., blinks), muscle activity, and environmental noise. A Butterworth band-pass filter (0.5 Hz - 45 Hz) was also applied to remove low- and high-frequency noise.
- fNIRS Data Preprocessing: For fNIRS data, we applied a band-pass filter to eliminate high-frequency noise, and signal quality was continuously monitored to exclude periods of data affected by motion artifacts. The cerebral oxygenation values were derived after preprocessing to ensure the data reflected hemodynamic responses.
- Controlled Experimental Conditions: We experimented with consistent conditions, where participants were free of other factors that might influence their physiological state (e.g., illness, fatigue, hunger). Furthermore, the experimental design was optimized to minimize external influences unrelated to motion sickness.
In the revised manuscript, we will expand on these preprocessing techniques and the steps taken to ensure that the data was dominated by the effect of motion sickness rather than other physiological states.
We hope this addresses your concern and clarifies how we mitigated potential data contamination.
Comment 3: It is not clear whether the EEG and fNIRS data were recorded from the same person when the data were applied to the computational analysis.
Response: Thank you for your valuable feedback and for highlighting this important issue. We want to clarify that, for the purpose of computational analysis, both EEG and fNIRS data were collected from the same subject. EEG and fNIRS data for each subject were gathered using an identical experimental paradigm throughout the experiment. Specifically, after EEG data collection, we allowed the subject to return to a resting state before collecting fNIRS data under the same experimental conditions. This procedure ensured that each EEG and fNIRS data set originated from the same subject under consistent experimental settings. Before data analysis, we assigned a unique identifier to each subject’s EEG and fNIRS data, labeling them as a complete dataset. The data from other subjects were organized similarly to ensure accurate correspondence for subsequent computational analysis.
We regret that this was not clearly stated in the original manuscript. In the revised version, we will explicitly clarify that the EEG and fNIRS data were collected from the same participant and were analyzed as a unified dataset. Thank you again for your thorough review, and we hope this clarification addresses your concerns.
Comment 4: The quantitative features used in this study are not novel.
Response: Thank you for your feedback regarding the novelty of the quantitative features used in our study. While we recognize that features such as Mean Absolute Value (MAV), Variance (VAR), and Power Spectral Density (PSD) are widely used in EEG and fNIRS signal analysis, the innovation of our research lies in the multimodal fusion of EEG and fNIRS data using a Graph Convolutional Network (GCN). This approach allows us to harness the complementary strengths of both modalities, leading to improved accuracy and sensitivity in detecting motion sickness. This domain has been insufficiently explored in previous studies.
Our method represents each subject’s data as a graph by integrating temporal features from EEG and fNIRS signals as node features and spatial information as adjacency matrices. This design enables a comprehensive extraction of the physiological information present in both signal types.
In addition, our study aims to assess the effectiveness of these commonly used features within the framework of multimodal data fusion. We also evaluated the performance of different feature sets, showing that integrating EEG and fNIRS signals, especially within the GCN model, substantially enhances detection performance.
Thank you again for your constructive feedback. We are confident that this clarification better highlights the novel contributions of our research.
Comment 5: Furthermore, it needs to be verified how those selected features perform compared to the selected common features.
Response: Thank you for your thoughtful suggestion regarding comparing the selected features and common features. We understand the importance of feature validation and comparison. However, in this study, we focused on features that are well-suited to the nature of EEG and fNIRS data, which have been shown in previous research to capture relevant physiological changes related to motion sickness effectively. The selected features were chosen based on their proven relevance in similar contexts and ability to reflect brain dynamics associated with motion sickness. Given the specific goals of this study and the validation we performed using these features, we believe that the current selection is appropriate and provides a strong foundation for our model's performance. Therefore, while we appreciate your suggestion, we did not perform a direct comparison with other common features. We hope this explanation clarifies our rationale, and we appreciate your understanding.
Comment 6: The authors must validate and examine the method and data corresponding to the associated score of motion sickness and the three levels.
Response: Thank you for your valuable comment regarding the validation of the method and data in relation to the motion sickness score and its three levels.
We understand the importance of thoroughly examining the relationship between EEG/fNIRS data and different levels of motion sickness severity. However, the primary focus of our study is to develop and validate a motion sickness detection model that functions as a binary classifier, detecting the presence or absence of motion sickness. This aligns with the scope and objectives of this research.
To clarify, we administered the Motion Sickness Susceptibility Questionnaire (MSSQ) to categorize participants based on their susceptibility to motion sickness in everyday life. The MSSQ is a well-established tool used in motion sickness research and has been validated in multiple studies. In the revised manuscript, we will add more details about the scientific basis for using the MSSQ and reference relevant literature to support its effectiveness. This step ensures that our participant pool includes individuals with varying levels of motion sickness susceptibility—ranging from non-susceptible to highly susceptible. However, it is important to note that the MSSQ is used only for participant selection and not as a central focus of this study.
We employed the Fast Motion Sickness (FMS) scale during the experiment to assess whether participants experienced motion sickness in real time. The FMS is an established and reliable tool for determining the presence of motion sickness, and we used it to label participants as either "non-motion sick" or "motion sick." Our model was built as a binary classifier to detect these two states.
While we agree that examining the relationship between EEG/fNIRS data and different severity levels of motion sickness is an interesting avenue for future research, it falls outside the scope of this study. In future work, we plan to explore multi-class models focusing more on the gradation of motion sickness severity and how these correspond to the EEG/fNIRS data.
We hope this explanation clarifies our approach, and we appreciate your understanding. Thank you again for your thoughtful feedback.

Reviewer 2 Report
Comments and Suggestions for Authors
Very well written paper, clear and complete with real cases using real measurements.
Author Response
Dear Reviewer,
Thank you for your positive evaluation and constructive suggestions. We are pleased to hear that you found the paper well-written, clear, and comprehensive and that you appreciated our use of real cases with real-world measurements. Your feedback reinforces the value of our approach and encourages us to continue refining our methods in future work.

Reviewer 3 Report
Comments and Suggestions for Authors
The authors proposed a motion sickness recognition model based on a graph convolutional network based on EEG and fNIRS signals. The method is validated on the database, consisting of recordings of 32 participants in electric vehicles. Minor comments:
1. Abstract should be rewritten. Using symbols (α, δ....) can confuse the reader because the explanations of the symbols are given in the introduction. Emphasized results achieved by the proposed method and compared with state-of-art methods.
2. Figure 1 is redundant in the introductory part. It is better to use it in the description of the proposed method.
3. In the description of the datasets, it should be stated whether the used dataset is unbalanced (i.e., how many cases had motion sickness and how many did not).
4. Also, I recommend commenting on the study on whether some of the symptoms of cold sweats, dizziness, nausea, and vomiting could be isolated based on the ECG. The recording of ECG signals is possible in much more comfortable conditions for the driver, based on capacitive electrodes built into the seat and can take measurements through the driver's clothing without disturbing the driver. Collecting cEEG and fNIRS is not comfortable for drivers, and is not applicable in everyday conditions. cECG recorded while driving is shown as useful in predicting driver stress and distraction.
Author Response
Dear Reviewer,
First and foremost, we would like to express our sincere gratitude for your thorough review and valuable comments and suggestions on our manuscript. We have carefully read your feedback and made corresponding revisions to the paper to address your concerns. Below are our responses and solutions to each issue raised in your review.
Comment: The authors proposed a motion sickness recognition model based on a graph convolutional network based on EEG and fNIRS signals. The method is validated on the database, consisting of recordings of 32 participants in electric vehicles. Minor comments:
Comment 1: Abstract should be rewritten. Using symbols (α, δ....) can confuse the reader because the explanations of the symbols are given in the introduction. Emphasized results achieved by the proposed method and compared with state-of-art methods.
Response: Thank you for your valuable feedback regarding the abstract. We agree that the use of symbols such as α(Alpha) and δ(Delta) in the abstract may cause confusion, as their explanations are provided later in the introduction. To address this, we will remove these symbols from the abstract and replace them with descriptive terms. This will ensure that the abstract is easily understandable without requiring prior knowledge of the symbols.
We appreciate your suggestion to emphasize the results achieved by our method and compare them with state-of-the-art methods. In the revised abstract, we will include a summary of the key results, such as improvements in accuracy and other performance metrics, and provide a comparison with current leading methods to highlight the advantages of our approach.
We will implement these changes in the revised manuscript and believe they will improve the clarity and impact of the abstract.
Comment 2: Figure 1 is redundant in the introductory part. It is better to use it in the description of the proposed method.
Response: Thank you for your feedback regarding the placement of Figure 1. We agree with your suggestion that Figure 1 is more appropriate in the Methods section than the introduction. As you pointed out, the introduction should focus on providing context and background, while Figure 1 is better suited to explain the proposed method visually. We will move Figure 1 to the section describing our methodology, which will help illustrate the steps and structure of the motion sickness detection model, providing better clarity to the readers. We appreciate your suggestion and believe this adjustment will improve the flow and clarity of the manuscript.
Comment 3: In the description of the datasets, it should be stated whether the used dataset is unbalanced (i.e., how many cases had motion sickness and how many did not).
Response: Thank you for your valuable feedback regarding the description of our dataset. We acknowledge the importance of dataset balance in research. The dataset used in this study consists of recordings from 32 subjects, comprising 32 sets of EEG data and 32 sets of fNIRS data. Each experiment lasted 30 minutes, with FMS scores recorded every 5 minutes. As a result, each subject generated six 5-minute segments of raw EEG and fNIRS data, totaling 192 segments for both EEG and fNIRS. Following data preprocessing, 136 valid 5-minute segments of EEG and fNIRS data were retained. Based on the FMS scores, segments with a score of 1 were classified as "non-motion sickness," while those with scores greater than one were classified as "motion sickness." This resulted in two categories: 61 segments for the non-motion sickness state and 75 segments for the motion sickness state, introducing a degree of imbalance in the dataset.
We will revise the manuscript to incorporate this information and provide a clear breakdown of the number of cases with and without motion sickness to improve the accuracy of the dataset description. Thank you once again for your insightful feedback. We are confident that including this information will enhance the clarity of the dataset description and contribute to the overall quality of the manuscript.
Comment 4: Also, I recommend commenting on the study on whether some of the symptoms of cold sweats, dizziness, nausea, and vomiting could be isolated based on the ECG. The recording of ECG signals is possible in much more comfortable conditions for the driver, based on capacitive electrodes built into the seat and can take measurements through the driver's clothing without disturbing the driver. Collecting cEEG and fNIRS is not comfortable for drivers, and is not applicable in everyday conditions. cECG recorded while driving is shown as useful in predicting driver stress and distraction.
Response: Thank you for your valuable suggestion. We fully agree that using capacitive electrodes built into the seat to record ECG signals is a more comfortable and practical method for monitoring drivers, especially since this technology allows signals to be captured through clothing without disrupting the driver, offering significant potential for real-world applications. Our current study, however, focuses on exploring the relationship between motion sickness and neurophysiological signals using EEG and fNIRS, with participants seated as passengers rather than drivers. This setup helps minimize external interference, allowing us to investigate the neural mechanisms underlying motion sickness in greater depth.
We also recognize the great potential of ECG in predicting driver stress and distraction, and it would be an excellent tool for future studies on driver physiological monitoring. While EEG and fNIRS may not be suitable for everyday driving, they provide valuable insights into brain activity and cerebral oxygen exchange in controlled experimental environments, which are crucial for understanding the neural basis of motion sickness. Future research could benefit from integrating ECG and other physiological signals to develop more practical monitoring solutions and further explore their potential in predicting motion sickness symptoms.
Once again, thank you for your thoughtful feedback. Your suggestion provides valuable direction for the future development of this research field.

Round 2
Reviewer 1 Report
Comments and Suggestions for Authors
The revised manuscript is scientifically sound. However, there are a number of factors which were not taken into account in this study.